# Association of urinary excretion rates of uric acid with biomarkers of kidney injury in patients with advanced chronic kidney disease

Antía López Iglesias[1], Marta Blanco Pardo[1], Catuxa Rodríguez Magariños[1], Sonia Pértega[2,3], Diego Sierra Castro[1], Teresa García Falcón[1], Ana Rodríguez-Carmona[1], Miguel Pérez Fontán[1,4]*

1 División of Nephrology, A Coruña University Hospital, A Coruña, Spain, 2 Rheumatology and Health Research Group, Faculty of Health Sciences, A Coruña University, A Coruña, Spain, 3 Nursing and Health Care Research Group, A Coruña Institute of Biomedical Reasearch (INIBIC), A Coruña, Spain, 4 Department of Medicine, Faculty of Health Sciences, A Coruña University, A Coruña, Spain

* miguel.perez.fontan@udc.es

**Data Availability Statement:** I confirm that all relevant data are within the manuscript and its Supporting information files.

## Abstract

### Background

The potential influence of hyperuricemia on the genesis and progression of chronic kidney disease (CKD) remains controversial. In general, the correlation between blood levels of uric acid (UA) and the rate of progression of CKD is considered to be modest, if any, and the results of relevant trials oriented to disclose the effect of urate-lowering therapies on this outcome have been disappointing. Urinary excretion rates of UA could reflect more accurately the potential consequences of urate-related kidney injury.

### Method

Using a cross-sectional design, we investigated the correlation between different estimators of the rates of urinary excretion of UA (total 24-hour excretion, mean urinary concentration, renal clearance and fractional excretion)(main study variables), on one side, and urinary levels of selected biomarkers of kidney injury and CKD progression (DKK3, KIM1, NGAL, interleukin 1b and MCP)(main outcome variables), in 120 patients with advanced CKD (mean glomerular filtration rate 21.5 mL/minute). We took into consideration essential demographic, clinical and analytic variables with a potential confounding effect on the explored correlations (control variables). Spearman's rho correlation and nonlinear generalized additive regression models (GAM) with p-splines smoothers were used for statistical analysis.

### Main results

Multivariate analysis disclosed independent correlations between urinary UA concentrations, clearances and fractional excretion rates (but not plasma UA or total 24-hour excretion rates of UA), on one side, and the scrutinized markers. These correlations were more consistent for DKK3 and NGAL than for the other biomarkers. Glomerular filtration rate,

**Funding:** The author(s) received no specific funding for this work.

**Competing interests:** The authors have declared that no competing interests exist.

proteinuria and treatment with statins or RAA axis antagonists were other independent correlates of the main outcome variables.

## Conclusions

Our results support the hypothesis that urinary excretion rates of UA may represent a more accurate marker of UA-related kidney injury than plasma levels of this metabolite, in patients with advanced stages of CKD. Further, longitudinal studies will be necessary, to disclose the clinical significance of these findings.

## Introduction

Hyperuricemia is a common metabolic disorder in humans. The threshold for the diagnosis of this condition varies somewhat in different studies, but 6.5–7.0 mg/dL is the most commonly accepted limit, based on the solubility of uric acid (UA) above which crystal deposition and gout are more likely [1]. However, the clinical consequences of hyperuricemia go beyond crystalline arthritis and urolithiasis, and the last decades have contemplated a persistent interest in the potential relationships between hyperuricemia, on one side, and arterial hypertension, diabetes, cardiovascular complications or kidney disease, on the other [2–5]. To which extent the links among UA levels and these conditions have a causal or comorbid nature remains controversial [6]. Crystalline UA is known to promote inflammation, oxidative stress and activation of the sympathetic nervous system and the renin-angiotensin-aldosterone axis [3, 4]. Potential consequences include endothelial injury, arterial rigidity, myocardial dysfunction, arterial hypertension and insulin resistance. For these reasons, hyperuricemia is seen by many as a definite cardiovascular risk factor. Urate-lowering therapy has been claimed to have beneficial effects on high blood pressure [7] and cardiovascular risk [8], although the available evidence is still inconclusive.

Hyperuricemia and chronic kidney disease (CKD) keep evident, yet complex interrelationships. On one side, both disorders coexist inside the metabolic-cardiovascular complex. In addition, CKD predisposes to hyperuricemia, an effect mediated by different mechanisms, which include a reduced glomerular filtration rate (GFR), genetic predisposition and the side effects of different medications for the management of CKD [9]. On the other side, UA crystal deposition is a well-known determinant of acute and persistent kidney injury [10, 11]. However, it is presently unclear if asymptomatic hyperuricemia can contribute to the generation and progression of CKD [9]. This controversial association could be a direct consequence of the above mentioned pathogenic effects of UA, and also be indirectly mediated by other factors (high blood pressure, medications). The current view is that asymptomatic hyperuricemia is, at most, a modest determinant of the progression of CKD [12–14]. This perception has been reinforced by the results of large clinical trials [15–17], indicating that UA lowering therapy does not appear to modify the course of this disorder.

The apparent paradox of a poor correlation between hyperuricemia and the progression of CKD, despite multiple putative mechanisms of UA-induced kidney injury, raises the question that the plasmatic level of UA may not be an accurate marker of this association. We have hypothesized that the rates of urinary excretion of UA may reflect more precisely the consequences of UA-related kidney damage, particularly in the presence of a reduced kidney mass. This alternative approach could be more effective at the time of disclosing local mechanisms of injury. We present the results of a cross-sectional study aimed to investigate the association

beween urinary excretion rates of UA and selected biomarkers of kidney damage, in a relatively large sample of patients with advanced CKD.

## Population and method

### General design and objectives

Following an observational, cross-sectional, single center design, we studied a group of patients with advanced CKD, with the main objective of disclosing potential associations between plasma levels and urinary excretion rates of UA, on one side, and selected clinical and biochemical markers of kidney injury, on the other.

The study complied with the essential ethical requirements requested for clinical observational studies, and the protocol was approved by the local Ethical Committee of the Coruña-Ferrol area hospitals (code 2016/381). Written informed consent was obtained from all patients participating in the study.

### Study population

For this study, we considered consecutive patients, incident or prevalent in the Advanced CKD Unit of our Division, under the following inclusion criteria:

- Age $\geq$ 18 years

- Estimated GFR (CKD-EPI) lower than 35 mL/minute at the time of recruitment

- General health condition permitting full participation in the study

- Full capacity and willingness to give informed consent and participate

- Exclusion criteria included:

- Failure to comply with any of the inclusion criteria.

- Active or recent ($<$12 months) renal replacement therapy (dialysis or kidney transplant).

- Symptomatic hiperuricemia, defined by active or recent ($<$12 months) episodes of gouty arthritis or uric acid lithiasis

- Any type of significant clinical event during the three months preceding recruitement

- Chronic infectious, malignant or inflammatory disease, active at any point during a six month period preceding recruitment.

- Inability, for any reason, to undergo body composition analysis or 24-hour ambulatory blood pressure records.

- Urinary incontinence or any other circumstance precluding complete 24 hour urine retrieval for the study.

Fig 1 displays the flow diagram of patient inclusión. Recruitment started in April 11, 2018, and was closed by May 7, 2019.

### Strategy of analysis

The main study variables were plasma UA levels and 24-hour urinary UA excretion rates, estimated in four different ways (see below). We intended to correlate these parameters with selected biomarkers of kidney tubular injury (outcome variables, see below). Our hypothesis was that urinary concentration or the amount of urinary excretion of UA could correlate better

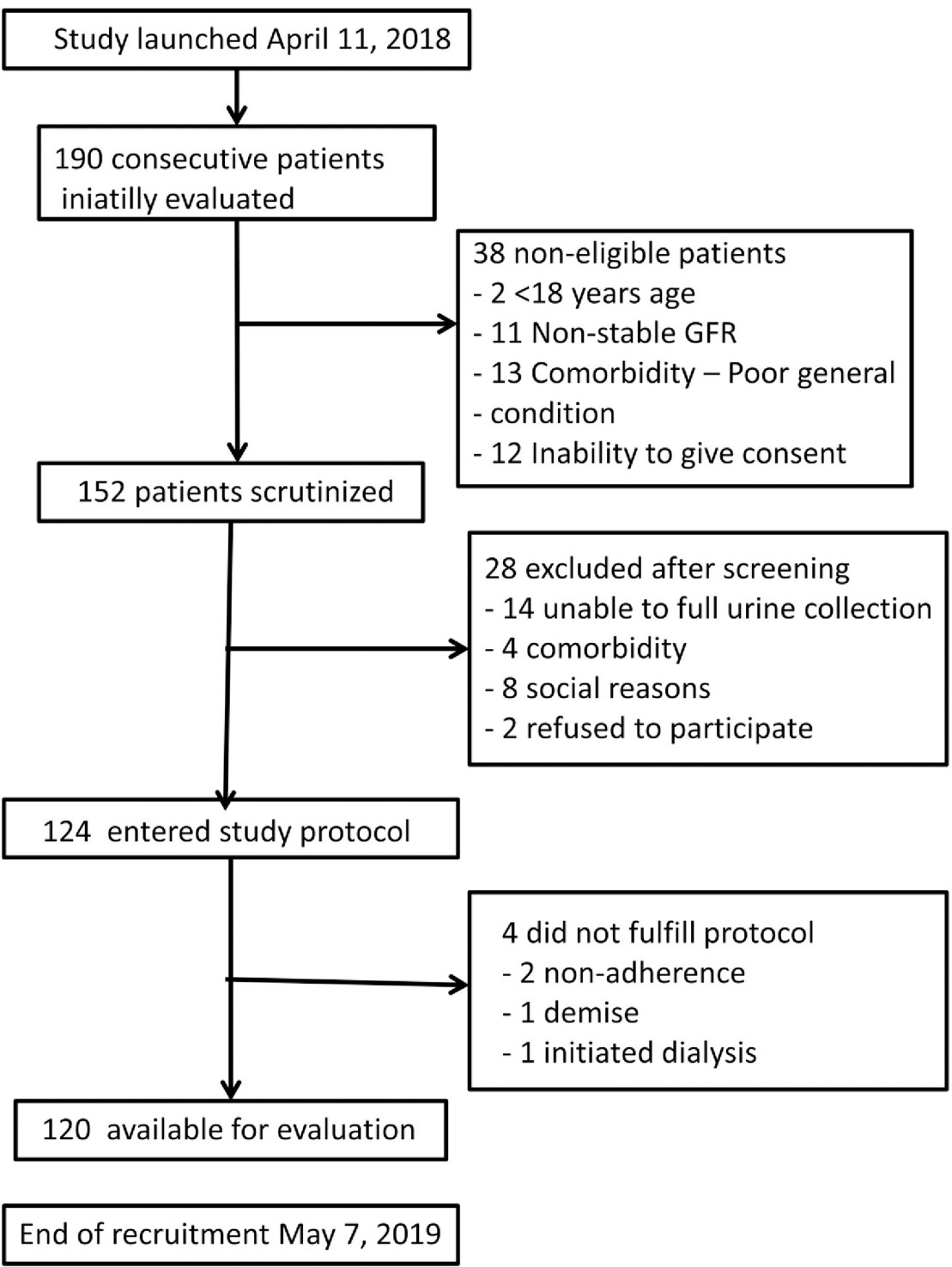

**Fig 1. Flow diagram of patient recruitment.**

with ongoing kidney injury than plasma UA concentration. This analysis could be biased by the well-known reduced capacity of UA excretion in the presence of a declining GFR. Considering that correction for GFR could better reflect UA-related injury to the remaining nephrons, this parameter was extensively used as a control variable. Physiopathologic considerations or preliminary analyses endorsed the analysis of other control variables (see below), to prevent or reduce confounding.

## Scrutinized variables

- Main study variables were: plasma UA concentration, and 24-hour urinary UA excretion rates, estimated as: mean urinary concentration (mg/dL), total excretion (mg/24 hours), kidney clearance (mL/minute), estimated from:

$$\text{Uric acid clearance}(\text{mL/min}) = [\text{urinary uric acid concentration}(\text{mg/dL})$$
$$* \text{24-hour diuresis}(\text{mL})/\text{plasma uric acid concentration }(\text{mg/dL}) * 1440],$$

and fractional excretion (FE), the latter estimated from:

$$\text{Uric FE} = 100 * [\text{mean urinary UA concentration}(\text{mg/dL})$$
$$* \text{plasma creatinine concentration}(\text{mg/dL})/\text{plasma UA concentration}(\text{mg/dL})$$
$$* \text{mean urinary creatinine concentration }(\text{mg/dL})]$$

- Main outcome variables included a set of presumed urinary markers of ongoing kidney injury: Dickkopf-related protein 3 (DKK3)(ELISA, Sigma-Aldrich, Merck, Darmstadt, GFR) (RAB0145), kidney injury molecule–1 (KIM-1)(ELISA, Enzo LIfe Sciences, Farmingdale, NY, USA)(ADI-900-226), neutrophil gelatinase-associated lipocalin (NGAL)(ELISA, Enzo LIfe Sciences)(BPD-KIT-036), Interleukin 1b (IL1b)(ELISA Quantikine, R&D Systems, Minneapolis, USA)(DLB50) and monocyte chemoattractant protein 1 (MCP) (ELISA Quantikine, R&D Systems, Minneapolis, USA)(DCP00). Urine samples were collected and frozen at -80°C until processing.

- Control variables included:

  - GFR, estimated from CKD-EPI (eGFR) and mean of urea and creatinine clearances (for clearance formulas see above)

  - 24-hour proteinuria

  - 24-hour urinary sodium and potassium excretion

  - Serum inflammatory and tissular damage markers [C-reactive protein (immunoturbidimetry), interleukin 6 (Immulite 2000, Siemens Diagnostics, Gwynedd, UK), and endothelin 1 (ELISA, Enzo LIfe Sciences)].

  - Demographic and clinical variables: age, gender, race, kidney disease, comorbidity (Charlson), diabetes, body mass index.

  - Body compositition (multifrequency bioimpedance analysis) parameters (lean and fat body mass, overhydration and ratio extracelular/intracelular wáter) and 24-hour ambulatory blood pressure records.

  - Laboratory variables: blood hemoglobin, plasma levels of urea, creatinine and albumin (autoanalyzer)

- Prescribed drugs, including erythropoietic agents, antihypertensives (renin-angiotensin axis RAA antagonists), urate- and lipid-lowering drugs, antiplatelet drugs, and diuretics.

## Statistical analysis

Data are presented as mean, standard deviation, quartiles or percentages for quantitative and categorical variables, respectively. The distribution of all examined variables was assessed by Kolmogorov-Smirnov test, histograms, and probability plots. A preliminary analysis showed a high [r > 0.95 (Pearson) in all cases, except MCP (r = 0.88)] degree of correlation between straight estimations of the urinary concentrations of biomarkers, on one side, and those generated after correcting for creatinine concentrations in urine, on the other, without significant differences in the results of the subsequent statistical analyses.

In the bivariate analysis, the relationship among UA levels and markers of kidney damage was assessed by nonparametric Spearman's rho correlation coefficient. Additionally, markers' values were compared among quartiles of different UA values, using the Kruskall-Wallis test to perform comparisons. Nonlinear generalized additive regression models (GAM) were then used to study the association of markers of kidney injury with plasma UA levels or UA excretion markers. Equations were adjusted, as needed, for potential confounders, including age, diabetes, GFR, proteinuria and RAA antagonist and statin therapies. For the non-linear relationships identified, smooth terms were introduced in the models using penalized regression splines. The validity of model assumptions was evaluated using analysis of residuals, and log-transformation of outcomes and independent variables were applied when necessary.

Scientific Package for Social Science (SPSS Statistics for Windows, Version 25.0, Chicago, IL, USA: SPSS Inc.) and R 3.6.3 (R Foundation for Statistical Computing, Vienna, Austria, accessed https://www.r-project.org/) software was used for statistical analyses. GAM models were adjusted using the 'mgdcv' package. A two-tailed value of p<0.05 was considered statistically significant. Data sets and synthaxis available at S1 and S2 Files. Main data analysis available at (1–3 Files).

## Results

Table 1 displays the main characteristics of the study population. Table 2 shows the results of the main laboratory blood/plasma variables scrutinized. Table 3 presents the urinary and clearance estimations. Remarkably, urinary biomarkers of kidney damage presented a skewed distribution (Fig 2), which made log transformation necessary for multivariate analysis.

Table 4 depicts the univariate correlations among urinary markers of kidney damage on one side, and the main variables scrutinized, on the other. Urinary NGAL and DKK3 concentrations kept the best correlation with UA excretion rates. We also observed different degrees of correlation among the different urinary markers of kidney damage, as also with serum interleukin 6. Finally, eGFR, proteinuria, statin therapy and ACEI-ARA therapy were other correlates of the outcome variables.

Multivariate analysis revealed a lack of association between plasma UA and the main outcome variables (Table 5, Fig 3). On the contrary, urinary UA excretion rates showed variable degrees of association with the outcome variables. This association was not apparent for total UA excretion (Table 6), but it was for urinary UA concentration (Table 7), UA clearance (Table 8) and fractional UA excretion (Table 9, Fig 4). Remarkably, most differences were significant only for the highest quartiles of the study variables (Tables 7–9), suggesting that only markedly elevated rates of UA excretion associate urinary markers of kidney injury. GFR and proteinuria were other direct, independent predictors of markers of kidney injury. On the

**Table 1. Study population.**

| | |
|---|---|
| Age (years) | 69.4 (11.4) |
| Gender (% males/females) | 74/46 (61.7/38.3) |
| Diabetes (%) | 41 (34.2) |
| Charlson's comorbidity score | 6.1 (2.0) |
| Body mass index (Kg/m$^2$) | 27.8 (4.7) |
| Lean body mass (Kg)* | 40.3 (9.9) |
| Fat mass (Kg)* | 23.9 (9.0) |
| Ratio intracellular/extracellular water* | 0.89 (0.10) |
| Overhydration (L)* | 0.82 (1.70) |
| Mean 24 hour systolic blood pressure (mm Hg)** | 138.9 (18.5) |
| Mean 24 hour diastolic blood pressure (mm Hg)** | 77.9 (8.6) |
| Antihypertensive therapy. Patients treated (%) | 110 (91.7) |
| Antihypertensive therapy. Number of drugs per patient | 2.3 (3.7) |
| RAA antagonists (%) | 61 (50.8) |
| Statin therapy (%) | 90 (75.0) |
| Allopurinol/Febuxostat (%) | 52 (42.5) |
| Loop diuretics (%) | 52 (42.5) |
| Antiplatelet drugs (%) | 57 (47.5) |
| EPO therapy (%) | 17 (14.2) |

*Multifrequency bioimpedance analysis.

**24-hour ambulatory pressure record.

Figures denote mean values (standard deviation)(numeric variables) or n (%)(categorized variables). RAA: Renin-angiotensin axis.

**Table 2. Laboratory values—Blood.**

| | |
|---|---|
| Hemoglobin (g/dL) | 12.4 (1.6) |
| Albumin (g/L) | 41.8 (3.2) |
| Creatinine (mg/dL) | 2.9 (0.9) |
| Urea (mg/dL) | 134.8 (50.2) |
| Uric acid (mg/dL) | 7.6 (1.7) |
| Sodium (mM/L) | 141.6 (2.5) |
| Potassium (mM/L) | 4.9 (1.9) |
| Endothelin 1 (pM) | 0.76 (1.35) |
| Ferritin (ng/dL) | 166.5 (164.0) |
| C reactive protein (mg/dL) | 0.24 (0.07–0.63) |
| Interleukin 6 (ng/mL) | 7.2 (4.8) |

Values denote mean (standard deviation) except C reactive protein (median with interquartile range)

contrary, statin therapy (but not RAA antagonists) associated lower levels of the main outcome variables.

## Discussion

UA is a weak organic acid generated primarily in the liver as an end-product of both endogenous (synthesis) and exogenous (diet) purine metabolism [6]. In contrast to other mammalian

**Table 3. Laboratory values—Urine and clearances.**

| | |
|---|---|
| Estimated glomerular filtration rate (CKD-EPI)(mL/minute) | 21.5 (6.4) |
| Diuresis (mL/24 hours) | 2020 (617) |
| Proteinuria (mg/24 hours) | 1219 (1629) |
| Creatinine clearance (mL/minute) | 28.1 (9.3) |
| Urea clearance (mL/minute) | 12.6 (4.7) |
| Mean of urea and creatinine clearances (mL/minute) | 20.3 (6.7) |
| Normalized protein nitrogen appearance (g/Kg/24 hours) | 1.09 (0.29) |
| Urinary pH* | 6.0 (5.0–6.5) |
| Urinary creatinine concentration (mg/dL) | 55.0 (17.3) |
| Urinary creatinine excretion (mg/24 hours) | 1054.7 (321.0) |
| Urinary sodium excretion (mM/24 hours) | 101.9 (48.9) |
| Urinary potassium excretion (mM/24 hours) | 63.6 (22.9) |
| Urinary uric acid excretion (mg/24 hours) | 277.3 (136.0) |
| Urinary uric acid concentration (mg/dL) | 14.0 (6.4) |
| Uric acid clearance (mL/minute) | 2.5 (1.2) |
| Fractional excretion of uric acid (%) | 9.9 (4.9) |
| Urinary DKK3 (ng/mL)* | 1.33 (0,38–2,18) |
| DKK3/Cr (ng/mg) | 2.39 (0.70–4.59) |
| Urinary NGAL (ng/mL)* | 15.22 (5.21–47.79) |
| NGAL/Cr (ng/mg) | 32.50 (9.86–98.91) |
| Urinary KIM-1 (pg/mL)* | 1110,3 (493,9–1963,8) |
| KIM-1/Cr (pg/mg) | 1919.81 (977.13–3763.86) |
| Urinary Il1B (pg/mL)* | 145,9 (79,5–213,8) |
| Il1b/Cr (pg/mg) | 274.52 (140.00–402.26) |
| Urinary MCP (pg/mL)* | 151,2 (88,1–281,7) |
| MCP/Cr (pg/mg) | 285.58 (180.46–490.25) |

Values denote mean (standard deviation) except those marked with * (median with interquartile range)

species, UA represents a terminal metabolic product in humans, its plasma levels being significantly higher than those observed in lower mammals. It is cleared from the body mainly by urinary excretion (approx. 2/3, or 600 mg/day, as a mean) and from the gastrointestinal tract (approx. 1/3). Its renal clearance keeps around 8–9 mL/minute, with a fractional excretion in a range between 7 and 10% [3, 6]. Kidney management of UA is rather complex, and includes glomerular filtration and tubular, transporter-driven reabsorption and secretion phases, taking place basically in the proximal nephron [6, 18].

Several potential pathogenic pathways support the hypothesis that excess UA levels may favor the appearance and progression of CKD. The most evident is the capacity of hyperuricemia to induce macro- (lithiasis) or micro-crystal (intratubular) precipitation [10, 11, 19, 20]. This circumstance may be directly detrimental for kidney function, mediated by urinary tract obstruction and, in the second case, inflammation, oxidative stress and activation of the sympathetic nervous system and the renin-angiotensin-aldosterone axis [3, 4]. On the contrary, soluble UA appears to bear antiinflammatory properties [9]. Hyperuricemia may also carry indirect effects on the course of CKD by impairing, for instance, blood pressure control [3, 21].

In contrast with the above mentioned considerations, previous studies have found a limited success at the time of translating the potential ability of asymptomatic hyperuricemia to induce kidney injury to clinical practice. In fact, the dominant, current view is that this disorder is, at

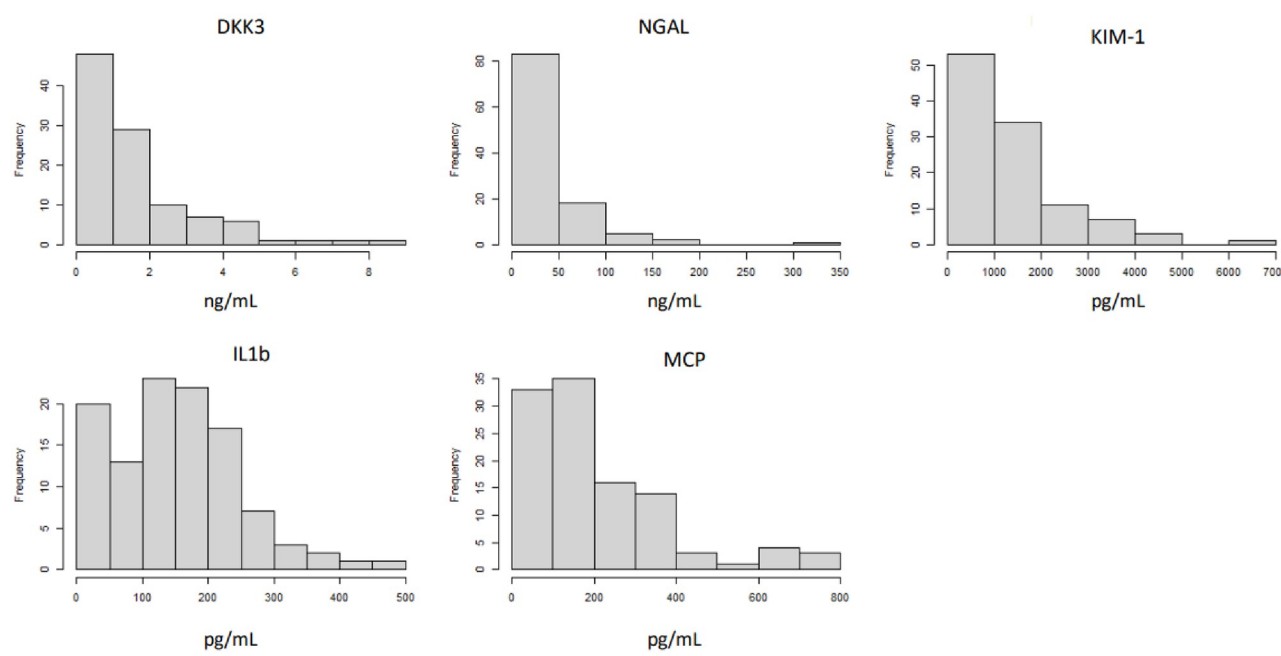

**Fig 2. Histogram of the distribution of the scrutinized markers of kidney injury.**

**Table 4. Main univariate correlations of the selected urinary markers*.**

| | DKK3/Cr urine | KIM-1/Cr urine | NGAL/Cr urine | IL1b/Cr urine | MCP/Cr urine |
|---|---|---|---|---|---|
| Plasma uric acid (mg/dL) | -0.07 (0.47) | 0.081 (0.40) | 0.073 (0.44) | 0.14 (0.15) | 0.14 (0.26) |
| Urinary uric acid excretion (mg/24 hours) | 0.11 (0.25) | 0.061 (0.62) | 0.17 (0.082) | 0.06 (0.52) | 0.014 (0.89) |
| Ratio uric acid excretion / eGFR (CKD-EPI) | 0.30 (0.002) | 0.15 (0.10) | 0.34 (0.0005) | -0.16 (0.09) | 0.11 (0.27) |
| Urinary uric acid concentration (mg/dL) | 0.12 (0.21) | 0.19 (0.051) | 0.13 (0.19) | 0.04 (0.33) | 0.09 (0.74) |
| Fractional excretion of uric acid (%) | 0.42 (0.0005) | 0.11 (0.25) | 0.34 (0.0005) | 0.016 (0.71) | 0.072 (0.46) |
| Uric acid clearance (mL/minute) | 0.18 (0.089) | 0.060 (0.54) | 0.19 (0.054) | 0.04 (0.80) | 0.15 (0.71) |
| DKK3/Cr urine (ng/mg) | - | 0.27 (0.005) | 0.40 (0.0005) | 0.28 (0.03) | 0.31 (0.001) |
| KIM-1/Cr urine (pg/mg) | 0.27 (0.005) | - | 0.48 (0.001) | 0.40 (0.001) | 0.65 (0.001) |
| NGAL/Cr urine (ng/mg) | 0.40 (0.0005) | 0.48 (0.001) | - | 0.54 (0.001) | 0.41 (0.003) |
| IL1b/Cr urine (pg/mg) | 0,28 (0.003) | 0.40 (0.001) | 0.54 (0.001) | - | 0.47 (0.001) |
| MCP/Cr urine (pg/mg) | 0.31 (0.001) | 0.65 (0.001) | 0.41 (0.003) | 0,47 (0.001) | - |
| Estimated GFR (CKD-EPI)(mL/minute) | -0.24 (0.013) | -0.07 (0.49) | -0.28 (0.003) | -0.20 (0.034) | -0.21 (0.025) |
| Proteinuria (mg/24 hours) | 0.36 (0.0005) | 0.28 (0.003) | 0.49 (0.0005) | 0.72 (0.001) | 0.36 (0.001) |
| Urinary pH | 0.31 (0.009) | -0.06 (0.70) | -0.01 (0.94) | -0.04 (0.79) | -0.03 (0.83) |
| Age (years) | 0.04 (0.66) | -0.077 (0.42) | -0.19 (0.043) | -0.18 (0.054) | -0,18 (0.064) |
| Plasma albumin (g/dL) | -0.07 (0.49) | -0.12 (0.19) | -0.18 (0.055) | -0.21 (0.029) | 0.29 (0.006) |
| Plasma endothelin 1 (pM) | 0.24 (0.016) | 0.067 (0.49) | 0.18 (0.076) | 0.06 (0.71) | 0.22 (0.020) |
| Plasma interleukin 6 (ng/mL) | 0.40 (0.0005) | 0.37 (0.002) | 0.44 (0.0005) | 0.09 (0.60) | 0.29 (0.04) |
| Statin therapy (ref. no) | -0.32 (0.009) | -0.19 (0.047) | -0.058 (0.55) | -0.039 (0.79) | -0.018 (0.058) |
| Loop diuretics (mg/24 hours) | -0.11 (0.23) | 0.07 (0.56) | -0.002 (0.98) | -0.022 (0.84) | 0.11 (0.27) |
| RAA antagonist therapy (ref. no) | 0.25 (0.010) | -0.085 (0.37) | -0.046 (0.63) | 0.02 (0.86) | -0.11 (0.25) |

Values denote Spearman's Rho correlation coefficients and associated p-vaues. Other correlations with variables in Tables 1–3 not significant.

**Table 5. Adjusted relationship between plasma uric acid levels and the selected urinary markers of kidney injury.**

| | DKK3(log) | | | NGAL(log) | | | KIM-1(log) | | | IL1b (log) | | | MCP(log) | | |
|---|---|---|---|---|---|---|---|---|---|---|---|---|---|---|---|
| | B | SE | P | B | SE | p | B | SE | p | B | SE | p | B | SE | P |
| (Intercept) | 0,329 | 0,469 | 0,485 | 1,611 | 0,357 | 0,000 | 3,064 | 0,291 | <2e-16 | 0,632 | 0,369 | 0,101 | 2,482 | 0,255 | 0,000 |
| **Plasma uric acid (ref. 1st quartile)** | | | | | | | | | | | | | | | |
| **2nd quartile** | -0,224 | 0,160 | 0,165 | -0,009 | 0,126 | 0,944 | -0,053 | 0,102 | 0,608 | 0,200 | 0,126 | 0,128 | -0,031 | 0,090 | 0,728 |
| **3rd quartile** | -0,052 | 0,172 | 0,762 | 0,096 | 0,134 | 0,474 | 0,077 | 0,109 | 0,482 | 0,039 | 0,134 | 0,776 | -0,071 | 0,095 | 0,457 |
| **4rd quartile** | -0,025 | 0,168 | 0,884 | 0,069 | 0,132 | 0,599 | -0,029 | 0,107 | 0,789 | 0,077 | 0,165 | 0,644 | -0,009 | 0,094 | 0,927 |
| Age (years) | 0,004 | 0,005 | 0,424 | 0,000 | 0,004 | 0,932 | 0,003 | 0,004 | 0,373 | 0,004 | 0,005 | 0,379 | -0,001 | 0,003 | 0,825 |
| GFR (mL/m) | -0,011 | 0,009 | 0,226 | -0,011 | 0,007 | 0,146 | -0,006 | 0,006 | 0,355 | -0,004 | 0,007 | 0,608 | -0,003 | 0,005 | 0,631 |
| Proteinuria (log) (mg/24 h) | | | **0,007***| 0,188 | 0,038 | **<0,001** | 0,096 | 0,031 | **0,003** | 0,041 | 0,043 | 0,351 | 0,101 | 0,027 | **<0,001** |
| Diabetes | -0,227 | 0,126 | 0,075 | 0,012 | 0,100 | 0,906 | 0,144 | 0,081 | 0,078 | 0,027 | 0,106 | 0,805 | 0,087 | 0,071 | 0,221 |
| Statin therapy | -0,357 | 0,139 | **0,012** | -0,090 | 0,111 | 0,422 | -0,198 | 0,090 | **0,031** | -0,023 | 0,113 | 0,840 | -0,161 | 0,079 | **0,045** |
| RAA antagonists | -0,106 | 0,117 | 0,366 | -0,011 | 0,093 | 0,909 | -0,063 | 0,076 | 0,411 | -0,206 | 0,089 | 0,031 | -0,034 | 0,066 | 0,608 |

Generalized additive regression (GAM) models adjusting for age, proteinuria (log-transformed), residual renal function, diabetes and statin or angiotensin-converting enzyme inhibitors treatment.

*Introduced as a non-linear tem in the model

GFR: Glomerular filtration rate; RAA: Renin-angiotensin axis

most, a modest determinant of the progression of CKD [12–14] and, as previously stated, large clinical trials [15–17] have not detected an effect of UA lowering therapy on the rate of decline of GFR in patients with CKD [3]. This apparent paradox raises the possibility that the plasma level of urate may not be a sensitive marker of UA-related kidney injury. Our hypothesis was that tubular concentrations of UA may reflect more accurately the consequences of UA-related damage. Increased tubular UA processing per functioning nephron in the setting of advanced CKD could compromise kidney function by several mechanisms, including direct tubular

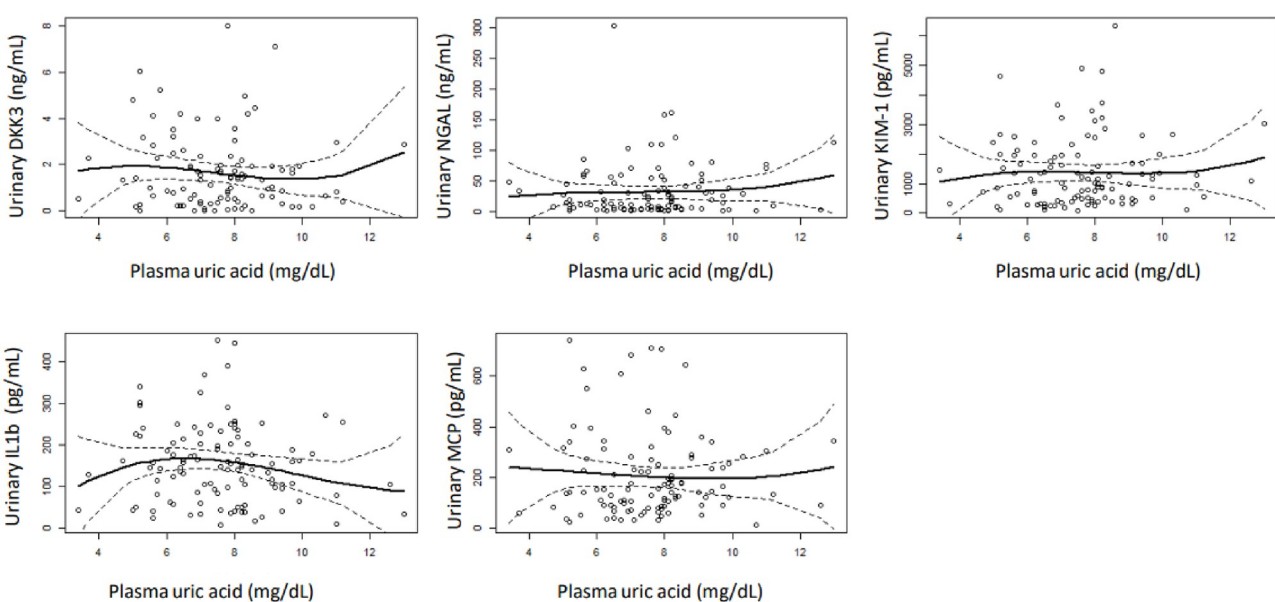

**Fig 3. Smoothing regression showing the relationship between plasma uric acid levels and the selected markers of kidney injury.**

**Table 6. Adjusted relationship between total urinary uric acid excretion (24 hours) and the selected urinary markers of kidney injury.**

| | DKK3(log) | | | NGAL(log) | | | KIM-1(log) | | | IL1b (log) | | | MCP(log) | | |
|---|---|---|---|---|---|---|---|---|---|---|---|---|---|---|---|
| | B | SE | P | B | SE | p | B | SE | p | B | SE | p | B | SE | P |
| (Intercept) | 0,230 | 0,445 | 0,606 | 1,596 | 0,177 | 0,000 | 3,085 | 0,283 | <0,001 | 0,980 | 0,375 | 0,016 | 2,414 | 0,249 | 0,000 |
| **Total urinary uric acid excretion (Ref. 1st quartile)** | | | | | | | | | | | | | | | |
| **2nd quartile** | 0,156 | 0,163 | 0,341 | -0,005 | 0,129 | 0,969 | 0,027 | 0,107 | 0,800 | 0,257 | 0,153 | 0,108 | 0,089 | 0,091 | 0,335 |
| **3rd quartile** | -0,003 | 0,162 | 0,985 | 0,124 | 0,128 | 0,332 | 0,079 | 0,106 | 0,459 | 0,049 | 0,137 | 0,722 | 0,035 | 0,091 | 0,699 |
| **4rd quartile** | 0,399 | 0,204 | 0,053 | 0,208 | 0,157 | 0,189 | -0,084 | 0,131 | 0,523 | 0,185 | 0,172 | 0,296 | -0,112 | 0,112 | 0,318 |
| Age (years) | 0,006 | 0,005 | 0,245 | | | 0,263 | 0,002 | 0,003 | 0,655 | 0,001 | 0,005 | 0,787 | -0,002 | 0,003 | 0,608 |
| GFR (mL/m) | -0,025 | 0,011 | **0,027** | -0,014 | 0,009 | 0,111 | -0,001 | 0,007 | 0,903 | -0,013 | 0,010 | 0,202 | 0,000 | 0,006 | 0,985 |
| Proteinuria (log) (mg/24 h) | | | **0,009*** | 0,184 | 0,038 | **<0,001** | 0,096 | 0,032 | **0,003** | 0,018 | 0,042 | 0,673 | | | **0,001*** |
| Diabetes | -0,217 | 0,122 | 0,079 | 0,000 | 0,098 | 0,998 | 0,139 | 0,080 | 0,086 | 0,026 | 0,087 | 0,767 | 0,105 | 0,068 | 0,128 |
| Statin therapy | -0,338 | 0,138 | **0,016** | -0,067 | 0,109 | 0,540 | -0,200 | 0,090 | **0,029** | -0,039 | 0,108 | 0,721 | -0,175 | 0,077 | **0,025** |
| RAA antagonists | -0,097 | 0,115 | 0,401 | -0,023 | 0,091 | 0,799 | -0,079 | 0,076 | 0,300 | -0,182 | 0,092 | 0,059 | -0,048 | 0,065 | 0,459 |

Generalized additive regression (GAM) models adjusting for age, proteinuria (log-transformed), residual renal function, diabetes and statin or angiotensin-converting enzyme inhibitors treatment.

*Introduced as a non-linear tem in the model

GFR: Glomerular filtration rate; RAA: Renin-angiotensin axis

toxicity, microcrystal precipitation or increased energy demand. Our results support, at least partly, this hypothesis. First, plasma levels of UA did not keep a correlation with markers of kidney injury (Table 5), which agrees with most of the available evidence. Neither did total urinary UA excretion correlate with the study outcome variables (Table 6). Any putative correlation of this parameter with biomarkers of kidney injury may be biased by the reduced overall urinary excretion of UA in the presence of a low GFR and an increased intestinal excretion of

**Table 7. Adjusted relationship between urinary uric acid concentration and the selected urinary markers of kidney injury.**

| | DKK3(log) | | | NGAL(log) | | | KIM-1(log) | | | IL1b (log) | | | MCP(log) | | |
|---|---|---|---|---|---|---|---|---|---|---|---|---|---|---|---|
| | B | SE | p | B | SE | p | B | SE | p | B | SE | p | B | SE | p |
| (Intercept) | 0,236 | 0,459 | 0,608 | 1,610 | 0,172 | <0,001 | 3,072 | 0,286 | <0,001 | 0,950 | 0,336 | 0,010 | 2,404 | 0,254 | 0,000 |
| **Urinary uric acid concentration (Ref. 1st quartile)** | | | | | | | | | | | | | | | |
| 2nd quartile | 0,247 | 0,158 | 0,121 | 0,097 | 0,121 | 0,423 | 0,196 | 0,102 | 0,057 | 0,078 | 0,117 | 0,510 | 0,140 | 0,090 | 0,124 |
| 3rd quartile | 0,263 | 0,155 | 0,094 | 0,039 | 0,120 | 0,749 | 0,041 | 0,100 | 0,684 | -0,114 | 0,122 | 0,358 | 0,060 | 0,089 | 0,503 |
| 4rd quartile | 0,397 | 0,185 | **0,035** | 0,369 | 0,142 | **0,011** | 0,217 | 0,119 | **0,061** | 0,278 | 0,118 | **0,053** | 0,107 | 0,105 | 0,310 |
| Age (years) | 0,004 | **0,005** | 0,496 | | | 0,266 | 0,002 | 0,003 | 0,481 | 0,003 | 0,004 | 0,503 | 0,000 | 0,003 | 0,975 |
| GFR (mL/m) | -0,019 | 0,010 | 0,056 | -0,016 | 0,008 | **0,046** | -0,008 | 0,006 | 0,196 | -0,017 | 0,008 | **0,043** | -0,006 | 0,006 | 0,320 |
| Proteinuria (log) (mg/24 h) | | | **0,011*** | 0,184 | 0,037 | **<0,001** | 0,091 | 0,031 | **0,004** | 0,037 | 0,035 | 0,300 | 0,097 | 0,027 | **0,001** |
| Diabetes | -0,256 | 0,126 | **0,045** | -0,052 | 0,098 | 0,594 | 0,085 | 0,081 | 0,299 | -0,070 | 0,085 | 0,417 | 0,072 | 0,072 | 0,318 |
| Statin therapy | -0,349 | 0,137 | **0,013** | -0,067 | 0,106 | 0,526 | -0,192 | 0,088 | **0,032** | 0,061 | 0,097 | 0,538 | -0,166 | 0,078 | **0,037** |
| RAA antagonists | -0,106 | 0,115 | 0,357 | -0,025 | 0,088 | 0,777 | -0,067 | 0,074 | 0,367 | -0,178 | 0,081 | **0,038** | -0,032 | 0,065 | 0,624 |

Generalized additive regression (GAM) models adjusting for age, proteinuria (log-transformed), residual renal function, diabetes and statin or angiotensin-converting enzyme inhibitors treatment.

*Introduced as a non-linear tem in the model

GFR: Glomerular filtration rate; RAA: Renin-angiotensin axis

**Table 8. Adjusted relationship between urinary uric acid clearance and the selected urinary markers of kidney injury.**

|  | DKK3(log) | | | NGAL(log) | | | KIM-1(log) | | | IL1b (log) | | | MCP(log) | | |
|---|---|---|---|---|---|---|---|---|---|---|---|---|---|---|---|
|  | B | SE | p | B | SE | p | B | SE | p | B | SE | p | B | SE | p |
| (Intercept) | 0,349 | 0,442 | 0,432 | 1,566 | 0,177 | <0,001 | 3,114 | 0,287 | 0,000 | 1,096 | 0,352 | 0,005 | 2,438 | 0,250 | 0,000 |
| **Uric acid clearance (Ref. 1st quartile)** |  |  |  |  |  |  |  |  |  |  |  |  |  |  |  |
| 2nd quartile | 0,148 | 0,155 | 0,341 | 0,125 | 0,125 | 0,317 | 0,053 | 0,105 | 0,612 | 0,158 | 0,120 | 0,201 | 0,023 | 0,092 | 0,799 |
| 3rd quartile | 0,130 | 0,188 | 0,492 | 0,216 | 0,141 | 0,129 | -0,027 | 0,120 | 0,822 | -0,178 | 0,190 | 0,361 | -0,007 | 0,104 | 0,949 |
| 4rd quartile | 0,513 | 0,189 | **0,008** | 0,295 | 0,147 | **0,048** | 0,048 | 0,124 | 0,701 | 0,187 | 0,152 | 0,233 | -0,035 | 0,108 | 0,747 |
| Age (years) | 0,005 | 0,005 | 0,373 |  |  | 0,249* | 0,002 | 0,003 | 0,550 | -0,001 | 0,005 | 0,834 | 0,000 | 0,003 | 0,881 |
| GFR (mL/m) | -0,028 | 0,011 | **0,010** | -0,017 | 0,008 | 0,051 | -0,005 | 0,007 | 0,453 | -0,010 | 0,009 | 0,248 | -0,002 | 0,006 | 0,720 |
| Proteinuria (log) (mg/24 h) |  |  | **0,029*** | 0,172 | 0,039 | **<0,001** | 0,100 | 0,033 | **0,003** | 0,035 | 0,041 | 0,393 | 0,103 | 0,029 | **0,001** |
| Diabetes | -0,167 | 0,121 | 0,169 | 0,037 | 0,097 | 0,699 | 0,143 | 0,080 | 0,076 | -0,004 | 0,081 | 0,959 | 0,093 | 0,070 | 0,186 |
| Statin therapy | -0,369 | 0,135 | **0,007** | -0,080 | 0,108 | 0,459 | -0,198 | 0,091 | **0,031** | 0,009 | 0,099 | 0,926 | -0,164 | 0,079 | **0,041** |
| RAA antagonists | -0,106 | 0,113 | 0,350 | -0,034 | 0,089 | 0,704 | -0,066 | 0,076 | 0,384 | -0,240 | 0,090 | 0,014 | -0,033 | 0,066 | 0,616 |

Generalized additive regression (GAM) models adjusting for age, proteinuria (log-transformed), residual renal function, diabetes and statin or angiotensin-converting enzyme inhibitors treatment.

*Introduced as a non-linear tem in the model

GFR: Glomerular filtration rate; RAA: Renin-angiotensin axis

urate usually observed in patients with advanced CKD [18]. On the contrary, urinary concentration (Table 7), clearance (Table 8) and fractional excretion (Table 9) of UA did sustain an independent correlation with the scrutinized outcomes, particularly in the case of urinary concentrations of DKK3 and NGAL.

For the present analysis, we scrutinized urinary biomarkers of ongoing kidney injury as surrogates of the risk of progression of CKD [22]. DKK3 is a stress-induced, tubular epithelia–derived, profibrotic glycoprotein that induces tubulointerstitial fibrosis through its action on

**Table 9. Adjusted relationship between fractional urinary excretion of uric acid clearance and the selected urinary markers of kidney injury.**

|  | DKK3 (log) | | | NGAL (log) | | | KIM-1 (log) | | | IL1b (log) | | | MCP (log) | | |
|---|---|---|---|---|---|---|---|---|---|---|---|---|---|---|---|
|  | B | SE | p | B | SE | p | B | SE | p | B | SE | p | B | SE | p |
| (Intercept) | -0,033 | 0,455 | 0,942 | 1,375 | 0,194 | 0,000 | 3,056 | 0,307 | 0,000 | 1,120 | 0,315 | 0,002 | 2,414 | 0,267 | 0,000 |
| **Fractional excretion of uric acid (Ref. 1st quartile)** |  |  |  |  |  |  |  |  |  |  |  |  |  |  |  |
| **2nd quartile** | 0,046 | 0,155 | 0,769 | 0,075 | 0,125 | 0,552 | 0,023 | 0,107 | 0,833 | -0,215 | 0,112 | 0,068 | 0,060 | 0,093 | 0,520 |
| **3rd quartile** | 0,443 | 0,160 | **0,007** | 0,220 | 0,129 | 0,093 | 0,008 | 0,111 | 0,941 | 0,006 | 0,101 | 0,955 | 0,028 | 0,096 | 0,770 |
| **4rd quartile** | 0,398 | 0,164 | **0,017** | 0,309 | 0,133 | **0,022** | 0,031 | 0,114 | 0,789 | 0,225 | 0,111 | **0,045** | -0,004 | 0,099 | 0,967 |
| Age (years) | 0,005 | 0,005 | 0,307 |  |  | 0,282* | 0,003 | 0,003 | 0,471 | 0,000 | 0,004 | 0,981 | 0,000 | 0,003 | 0,966 |
| GFR (mL/m) | -0,013 | 0,009 | 0,148 | -0,007 | 0,007 | 0,316 | -0,004 | 0,006 | 0,527 | -0,010 | 0,006 | 0,079 | -0,004 | 0,005 | 0,459 |
| Proteinuria (log) (mg/24 h) |  |  | **0,023*** | 0,172 | 0,038 | **0,000** | 0,096 | 0,032 | **0,004** | 0,001 | 0,033 | 0,975 | 0,103 | 0,028 | 0,000 |
| Diabetes | -0,172 | 0,120 | 0,155 | 0,043 | 0,096 | 0,658 | 0,135 | 0,081 | 0,101 | 0,065 | 0,074 | 0,389 | 0,081 | 0,071 | 0,256 |
| Statin therapy | -0,322 | 0,133 | **0,018** | -0,069 | 0,108 | 0,523 | -0,194 | 0,092 | **0,038** | -0,050 | 0,085 | 0,567 | -0,157 | 0,080 | 0,053 |
| RAA antagonists | -0,121 | 0,112 | 0,280 | -0,030 | 0,089 | 0,735 | -0,067 | 0,077 | 0,387 | -0,213 | 0,076 | **0,011** | -0,026 | 0,067 | 0,693 |

Generalized additive regression (GAM) models adjusting for age, proteinuria (log-transformed), residual renal function, diabetes and statin or angiotensin-converting enzyme inhibitors treatment.

*Introduced as a non-linear tem in the model

GFR: Glomerular filtration rate; RAA: Renin-angiotensin axis

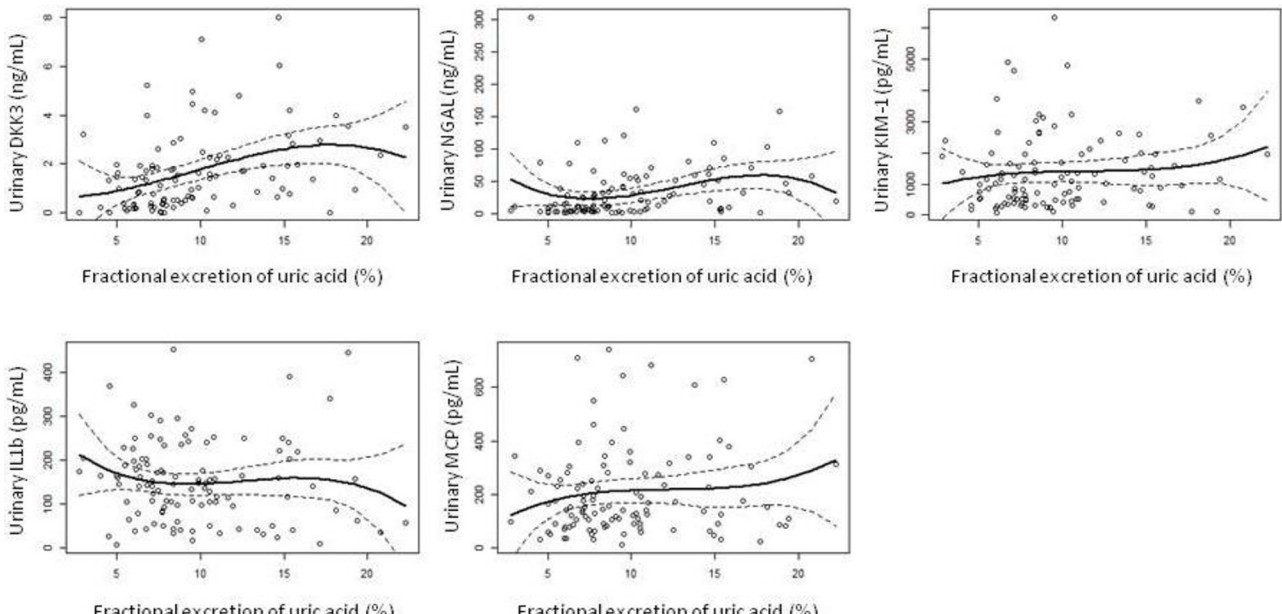

**Fig 4. Smoothing regression showing the relationship between fractional excretion of uric acid and the selected markers of kidney injury.**

the canonical Wnt/b-catenin signaling pathway [23, 24]. Moreover, DKK3 promotes renal fibrosis by promoting EMT, impairing angiogenic competence, activating TGF-β, and modulating local T cell responses [24]. Urinary DKK3 concentration has been consistently associated with the rate of decline of GFR in patients with CKD [24, 25].

NGAL is a member of the lipocalin superfamily, with a role as an innate antibacterial factor. However, this small protein is produced by many families of cells, including kidney tubular cells. Both inflammatory settings and proteinuria associate increased urinary levels of NGAL [26]. NGAL was seen for some time as a marker of acute, ongoing kidney injury, but more recent studies [22, 27–29] have disclosed a correlation with the severity and progression of CKD. On the other hand, expression of KIM-1 in renal epithelial cells leads to progressive interstitial kidney inflammation and fibrosis in rodent models, and is therefore supposed to have an unfavorable effect in CKD [30]. KIM-1 expression is upregulated in cases of ischemia, hypoxia, and cellular tubular injury and has been implicated in biologic mechanisms of CKD in the setting of diabetes mellitus [22]. Both urinary NGAL and KIM1 levels have been claimed to associate with chronic tubulonterstitial fibrosis and progression of CKD, but it is presently unclear if they have a pathogenic role (by promoting tubular cell apoptosis) or, alternatively, they perform as simple markers of ongoing damage. Two recent metaanalyses [22, 28] have suggested that urinary excretion of NGAL may represent a relatively accurate marker of kidney injury and, secondarily, progression of CKD. In the case of KIM1, the results are not equally consistent, as the above cited metaanalyses have disagreed at the time of confirming [22] or discarding [28] such a correlation. On the other hand, a relevant study [31] disclosed a significant association between urinary MCP1 levels, on one side, and the time course of CKD, although a metaanalysis could not confirm such association [22].

To our knowledge, only one previous study has addressed this question under a similar approach. Using a cross-sectional design, Zheng et al [32] observed an inverse correlation between 24-hour urinary UA excretion rates, on one side, and several tubular injury biomarkers, on the other. These results are in contrast with an absence of correlation observed in our

study (Table 6). The reasons for this discrepancy are not clear, but the different biomarkers scrutinized, some variations in the methodologic and statistical approach and, particularly, the level of CKD (mainly stages 2 and 3 in Zheng's study) and the amount of proteinuria, could have some influence on the differences observed.

Our study suffers from some significant limitations. The cross-sectional design did not permit to explore the correlation between markers of urinary excretion of UA and the rate of decline of GFR. This alternative, more direct approach was not undertaken due to a predictable lack of statistical power to detect minor expected effects. The significance of the selected biomarkers is questionable, as shown by the inconsistent correlations found for some of the parameters. The dependence of the selected variables on factors such as GFR, proteinuria or concomitant drug therapies may have generated some confounding at the time of data analysis. On the other hand, some outcome biomarkers may show a significant day to day variability [33], while others (as in the case of Il1b) may have an urologic rather than renal origin [34]. On the opposite view, among the strengths of the study, we should underline the careful, comprehensive design, with rather complete sets of study, coutcome and control variables.

In conclusion, urinary concentration of urate shows a significant, consistent association with selected biomarkers of progression of CKD (particularly urinary levels of DKK3 and NGAL), among patients with avanced CKD. Urinary excretion of UA may more sensitive than plasma levels of this solute, at the time of disclosing a potential correlation with the time course of CKD.

## Supporting information

**S1 File.**
(SPO)

**S2 File.**
(DOCX)

**S3 File.**
(DOCX)

**S4 File.**
(SAV)

**S5 File.**
(SPS)

**S1 Checklist. Human participants research checklist.**
(DOCX)

## Author Contributions

**Conceptualization:** Antía López Iglesias, Marta Blanco Pardo, Catuxa Rodríguez Magariños, Sonia Pértega, Diego Sierra Castro, Teresa García Falcón, Ana Rodríguez-Carmona, Miguel Pérez Fontán.

**Data curation:** Antía López Iglesias, Marta Blanco Pardo, Catuxa Rodríguez Magariños, Sonia Pértega, Diego Sierra Castro, Miguel Pérez Fontán.

**Formal analysis:** Sonia Pértega, Ana Rodríguez-Carmona, Miguel Pérez Fontán.

**Investigation:** Antía López Iglesias, Miguel Pérez Fontán.

**Methodology:** Antía López Iglesias, Marta Blanco Pardo, Catuxa Rodríguez Magariños, Sonia Pértega, Diego Sierra Castro, Miguel Pérez Fontán.

**Software:** Sonia Pértega.

**Validation:** Antía López Iglesias, Miguel Pérez Fontán.

**Writing – original draft:** Antía López Iglesias, Sonia Pértega, Teresa García Falcón, Ana Rodríguez-Carmona, Miguel Pérez Fontán.

**Writing – review & editing:** Sonia Pértega, Teresa García Falcón, Ana Rodríguez-Carmona, Miguel Pérez Fontán.

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
