## [Decision Letter · Decision Letter 0]

26 Mar 2024

PONE-D-24-04504Association of Urinary Excretion Rates of Uric Acid with Biomarkers of Kidney Injury in Patients with Advanced Chronic Kidney DiseasePLOS ONE

Dear Dr. Pérez Fontán,

Thank you for submitting your manuscript to PLOS ONE. After careful consideration, we feel that it has merit but does not fully meet PLOS ONE’s publication criteria as it currently stands. Therefore, we invite you to submit a revised version of the manuscript that addresses the points raised during the review process.

Your manuscript was reviewed by two experts and both of them suggested minor revision. Please address their comments as appropriate.

We look forward to receiving your revised manuscript.

Kind regards,

Partha Mukhopadhyay, Ph.D.

Section Editor

PLOS ONE

Reviewers' comments:

Reviewer's Responses to Questions

**Comments to the Author**

1. Is the manuscript technically sound, and do the data support the conclusions?

Reviewer #1: Yes

Reviewer #2: Yes

2. Has the statistical analysis been performed appropriately and rigorously? 

Reviewer #1: Yes

Reviewer #2: Yes

3. Have the authors made all data underlying the findings in their manuscript fully available?

Reviewer #1: Yes

Reviewer #2: Yes

4. Is the manuscript presented in an intelligible fashion and written in standard English?

Reviewer #1: Yes

Reviewer #2: Yes

5. Review Comments to the Author

Reviewer #1: Authors evaluated the relation of excreted uric acid with some kidney injury biomarkers in order to use such parameter as a marker of progression of kidney disease. Data is solid and well presented. I would ask to check the journal directives for presenting the paper.

The manuscript is overall well written, but with some typos, which makes me recommend a detailed review on the text to double-check if there are more mistakes other than the ones listed bellow:

- Along the introduction, you wrote Crystalline with an I instead of y.

- On the fifth paragraph of the discussion, you wrote progression in Spanish.

- On the seventh paragraph of discussion, you wrote questionable with double n.

- In the same paragraph, you bring a word cae which I believe should be case.

- The last phrase of the last paragraph needs to be rewritten. Some words might be missing.

- In reference 8, allopurinol is written with a single letter l.

Reviewer #2: The authors address the apparent paradox of a weak correlation between hyperuricemia and the progression of chronic kidney disease (CKD) by seeking an evaluation system that better reflects the roles of uric acid (UA) in CKD pathogenesis. Their findings suggest that urinary excretion rates of UA may serve as a more precise marker for identifying UA-related kidney injury.

Major Comments:

1. The results from tables 7 to 9 indicate significant differences primarily in the 4th quartile, suggesting that only extremely high urinary uric acid concentrations correlate well with urinary markers of kidney injury. However, the presence of other symptoms due to elevated urinary uric acid levels may confound the assessment of UA-related kidney damage. Hence, it is crucial to ascertain whether the authors included criteria for asymptomatic hyperuricemia in patient recruitment. If so, clarification regarding these criteria is warranted.

2. It appears that Figure 3 and Figure 4 are identical except for the x-axis title.

3. Could the authors provide the urinary creatinine levels? Since all scrutinized kidney injury markers are normalized by creatinine, this information is essential for contextualizing the findings.

4. The authors should provide more descriptive interpretations of their results rather than solely presenting statistical analysis tables in the manuscript.

Minor Comments:

1. Page 2, line 5, "resuts" should be "results."

2. Page 3, line 13, "hiperuricemia" should be "hyperuricemia."

3. From table 5 to table 9, "4rd quartile" should be "4th quartile."

4. Please ensure consistency in number formatting across different tables; some numbers use ".", while others use ",".

5. Please provide the catalogue numbers of the ELISA kits used in the manuscript.

6. In the methods section, the authors should include the formulas used for calculating eGFR, creatinine clearance, urea clearance, and uric acid clearance, respectively.

6. PLOS authors have the option to publish the peer review history of their article (what does this mean?). If published, this will include your full peer review and any attached files.

Reviewer #1: No

Reviewer #2: No

---

## [Author Response · Author response to Decision Letter 0]

18 Apr 2024

Dear Editor,

Thank you for your positive evaluation of our manuscript “Association of urinary excretion rates of uric acid with biomarkers of kidney injury in patients with advanced chronic kidney disease” (PONE D-24-4504). We are submitting a revised version of the paper, in which we have done our best to follow the recommendations of the reviewers and the Editorial Board. We hope that this new version may be satisfactory, but will consider with pleasure any additional request.

We include a marked-up manuscript, with the modifications highlighted in red, as also an unmarked version of the paper. We have reviewed the manuscript, looking for typographic errors. We have no further allegations concerning the Journal requirements. Please, let us know if there is still something wrong or equivocal. 

We shall answer now in detail to the requests o the reviewers:

 Reviewer #1

We thank the reviewer for his/her kind comments. We have made the six text corrections suggested

Reviewer #2

Again, thank you for your kind comments.

- We have included a paragraph in the Method chapter, to define asymptomatic hiperuricemia. Patients with active or recent (<12 months) gouty arthritis or uric litiasis were excluded (n=2 patients, included in Figure 1 as “comorbidity” cases)

- We have underlined in Results the fact that only 4th quartile levels of UA excretion showed a consistent association with markers of injury

- Yes, the reviewer is right!. Figures 3 and 4 are the same. We include now real Figure 4. We apologize for the mistake

- We include now urinary creatinine level (Table 3)

- We have extended our comments in the Results although, in general, we preferred to leave this for the Discussion

- We have corrected the indicated typographic errors

- We have improved the consistency of the figures, making pertinent changes (Tables 5 to 9)

- We include now the codes of the ELISA kits

- We have added the formulas for estimation of urea, creatinine and UA clearances.

Again, thank you for your help. We await Word from your ditorial Board. Any new request will be considered.

Sincerely

Miguel Pérez Fontán

Corresponding author

---

## [Decision Letter · Decision Letter 1]

7 May 2024

Association of Urinary Excretion Rates of Uric Acid with Biomarkers of Kidney Injury in Patients with Advanced Chronic Kidney Disease

PONE-D-24-04504R1

Dear Dr. Pérez Fontán,

We’re pleased to inform you that your manuscript has been judged scientifically suitable for publication and will be formally accepted for publication once it meets all outstanding technical requirements.

Kind regards,

Partha Mukhopadhyay, Ph.D.

Section Editor

PLOS ONE

Additional Editor Comments (optional):

Reviewers' comments:

Reviewer's Responses to Questions

**Comments to the Author**

1. If the authors have adequately addressed your comments raised in a previous round of review and you feel that this manuscript is now acceptable for publication, you may indicate that here to bypass the “Comments to the Author” section, enter your conflict of interest statement in the “Confidential to Editor” section, and submit your "Accept" recommendation.

Reviewer #1: All comments have been addressed

Reviewer #2: All comments have been addressed

2. Is the manuscript technically sound, and do the data support the conclusions?

Reviewer #1: Yes

Reviewer #2: Yes

3. Has the statistical analysis been performed appropriately and rigorously? 

Reviewer #1: Yes

Reviewer #2: Yes

4. Have the authors made all data underlying the findings in their manuscript fully available?

Reviewer #1: Yes

Reviewer #2: Yes

5. Is the manuscript presented in an intelligible fashion and written in standard English?

Reviewer #1: Yes

Reviewer #2: Yes

6. Review Comments to the Author

Reviewer #1: Authors responded to all requests in a satisfactory fashion, and I recommend the publication of this manuscript.

Reviewer #2: Please ensure consistency in number formatting in table 3; some numbers use ".", while others use ",".

7. PLOS authors have the option to publish the peer review history of their article (what does this mean?). If published, this will include your full peer review and any attached files.

Reviewer #1: **Yes: **Bruno Paes Leme Ferreira

Reviewer #2: No

---

## [Editor Report · Acceptance letter]

17 May 2024

PONE-D-24-04504R1 

PLOS ONE

Dear Dr. Pérez Fontán, 

I'm pleased to inform you that your manuscript has been deemed suitable for publication in PLOS ONE. Congratulations! Your manuscript is now being handed over to our production team.

Kind regards, 

on behalf of

Dr. Partha Mukhopadhyay 

Section Editor

PLOS ONE